# Cats shedding pathogenic *Leptospira* spp.—An underestimated zoonotic risk?

**Roswitha Dorsch**[1]*, **Javier Ojeda**[2], **Miguel Salgado**[3], **Gustavo Monti**[3], **Bernadita Collado**[3], **Camillo Tomckowiack**[4], **Carlos Tejeda**[3], **Ananda Müller**[2¤], **Theo Eberhard**[5], **Henricus L. B. M. Klaasen**[6], **Katrin Hartmann**[1]

**1** Medizinische Kleintierklinik, Ludwig-Maximilians-Universität Munich, Munich, Germany, **2** Instituto de Ciencias Clínicas, Facultad de Ciencias Veterinarias, Universidad Austral de Chile, Valdivia, Chile, **3** Instituto de Medicina Preventiva Veterinaria, Facultad de Ciencias Veterinarias, Universidad Austral de Chile, Valdivia, Chile, **4** Escuela de Graduados, Facultad de Ciencias Veterinarias, Universidad Austral de Chile, Valdivia, Chile, **5** University of Applied Sciences Munich, München, Germany, **6** Department of Global Companion Animals R&D, MSD Animal Health, Boxmeer, The Netherlands

¤ Current address: Department of Biomedical Sciences, Ross University School of Veterinary Medicine, Saint Kitts and Nevis, West Indies

* roswitha.dorsch@lmu.de

**Data Availability Statement:** All relevant data are within the manuscript and its Supporting Information files.

## Abstract

Shedding of DNA of pathogenic *Leptospira* spp. has been documented in naturally infected cats in several countries, but urinary shedding of infectious *Leptospira* spp. has only recently been proven. The climate in Southern Chile is temperate rainy with high annual precipitations which represents ideal preconditions for survival of *Leptospira* spp., especially during spring and summer. The aims of this study were to investigate shedding of pathogenic *Leptospira* spp. in outdoor cats in Southern Chile, to perform molecular characterization of isolates growing in culture, and to assess potential risk factors associated with shedding. Urine samples of 231 outdoor cats from rural and urban areas in southern Chile were collected. Urine samples were investigated for pathogenic *Leptospira* spp. by 4 techniques: qPCR targeting the *lipL32* gene, immunomagnetic separation (IMS)-coupled qPCR (IMS-qPCR), direct culture and IMS-coupled culture. Positive urine cultures were additionally confirmed by PCR. Multilocus sequence typing (MLST) was used to molecularly characterize isolates obtained from positive cultures. Overall, 36 urine samples (15.6%, 95% confidence interval (CI) 11.4–20.9) showed positive results. Eighteen (7.8%, 95% CI 4.9–12.1), 30 (13%, 95% CI 9.2–18), 3 (1.3%, 0.3–3.9) and 4 cats (1.7%; 95% CI 0.5–4.5) were positive in qPCR, IMS-qPCR, conventional culture, and IMS-coupled culture, respectively. MLST results of 7 culture-positive cats revealed sequences that could be assigned to sequence type 17 (6 cats) and sequence type 27 (1 cat) corresponding to *L. interrogans (Pathogenic Leptospira Subgroup 1)*. Shedding of pathogenic *Leptospira* spp. by cats might be an underestimated source of infection for other species including humans. The present study is the first one reporting growth of leptospires from feline urine in culture in naturally infected cats in South-America and characterisation of culture-derived isolates. So far, very few cases of successful attempts to culture leptospires from naturally infected cats are described worldwide.

**Funding:** The Bavarian Academic Center for Latin America supported the study by paying for travel expenses (between Valdivia and Munich on two occasions) for planning the study. The Dirección de Investigación y Desarollo of the Universidad Austral de Chile supported the study by covering part of the costs for the material necessary for sampling and part of additional costs, e.g. transport and compensation of owners for study participation. For compensation, all included cats were vaccinated and dewormed. MSD Animal health provided the resources for sampling, clinical investigation of cats and laboratory investigations. Part of the laboratory work was funded by Proyecto FIC 15-8 (código BIP 30421496) del Gobierno Regional de Los Ríos, Chile.

**Competing interests:** H.L.B.M. Klaasen is an employee of a company (MSD Animal Health) developing and marketing animal vaccines and pharmaceuticals. Otherwise the authors declare that they have no conflict of interest with respect to the research, authorship and/or publication of this article. This does not alter our adherence to PLOS ONE policies on sharing data and materials.

## Introduction

Leptospirosis is a worldwide zoonotic disease affecting most mammalian species. Prevalence of *Leptospira* species (spp.) infection is associated with certain environmental conditions and shows a positive association with increased rainfall and temperatures [1–4]. Leptospires survive for months in water and moist soil. Contamination of the environment with pathogenic *Leptospira* spp. mainly occurs via urinary shedding of reservoir hosts, such as small rodents that are considered the most important reservoir species. These species usually do not show clinical signs after infection but harbor leptospires in their renal tubules and shed them for prolonged periods of time [5]. In livestock animals (cows and pigs), leptospirosis causes chronic infections affecting fertility and reproduction [6, 7].

Only few reports of clinical cases of leptospirosis in cats are published [8–10], and *Leptospira* spp. infection appears to be underinvestigated in cats compared to other animal species. It has, however, been demonstrated that cats can be infected and develop specific antibodies [11, 12]. Reported antibody prevalence in naturally infected cats ranges from 0.8% to 33.3%, depending on the geographic area and the cats´ lifestyle [13–16]. The highest prevalence of antibody-positive cats have been found in Greece (33.3%) [17] and in rural areas of Southern Chile (25.2%) [18]. In addition, there are reports that infection with pathogenic *Leptospira* spp. in cats might be, although rarely, associated with acute kidney injury or liver disease but also with the development of chronic kidney disease. A study from Canada identified a significantly higher antibody prevalence in cats with chronic kidney disease (14.9%) when compared to healthy cats (7.2%) [12].

Still there are uncertainties concerning the role of cats as carrier of *Leptospira* spp. and their potential zoonotic risk. Few studies demonstrated shedding of *Leptospira* spp. in cats, first by identification of leptospiral agglutinins in the urine [19] and later by PCR [11, 12, 14–16, 20]. Experimentally infected cats can shed leptospires via urine intermittently up to 10 weeks after infection with *Leptospira interrogans* serovar Canicola [19]. Prevalence of cats shedding pathogenic *Leptospira* spp. in naturally infected cats has also been documented using urine PCR, ranging from 0.8% in Thailand [15], 1.7% in Spain [16], 3.3% in Canada and Germany [12, 14], 4.9% in Malaysia [20], up to 67.8% in Taiwan [11].

The climate in Southern Chile is temperate rainy with an average temperature of 16.7˚C. In the Los Rios region, the reported annual precipitations exceed 2400 mm (www.wetterkontor. de). This represents ideal environmental conditions for survival of pathogenic *Leptospira* spp. in the environment, particularly during spring and summer. A study performed in rural communities in Southern Chile demonstrated the ubiquity of leptospires in household environments [21]. Water samples from puddles, containers, animal troughs, rivers, canals, and drinking water from 236 households were investigated for the presence of pathogenic *Leptospira* spp. using PCR. Pathogenic *Leptospira* spp. were identified in 13.5% of all samples; e.g. in 11.3% of open containers, 14.5% of animal drinking sources, 15.8% of human drinking sources, and 19.3% of puddles. This indicates that there is a considerable risk of infection for humans and mammalian animals living in this environment. Leptospirosis has emerged to become a major public health threat in Latin America. Nevertheless, little is currently known regarding its incidence [22]. In countries, such as Peru and Chile, where leptospirosis has been studied systematically in rural areas, a large proportion of the human population has antibodies indicating previous exposure to pathogenic leptospires (36,6% and 12,0% respectively) [23, 24].

In Chile, a variable human to cat ratio ranging from 9.3:1 to 1.4:1 has been reported [25] suggesting that there is a relatively large feline population that could be a potential source of *Leptospira* spp. infection. So far, it is not known how many cats in this geographic area actively

shed pathogenic *Leptospira* spp., while the reported antibody prevalence in cats in rural areas (25.2%) [18] is among the highest reported worldwide. Therefore, the role of cats as a reservoir for pathogenic *Leptospira* spp. could be very important.

Thus, to provide scientific knowledge for a better understanding of the role and relative importance of cats as a source of pathogenic *Leptospira* spp. infection, the aims of the study were: (1) to determine the proportion of cats shedding pathogenic *Leptospira* spp. in Southern Chile, (2) to determine the molecular profile of the cultured isolates, and (3) to identify possible risk factors regarding *Leptospira* spp. shedding.

## Material and methods

### Ethical note

Procedures were approved by the Universidad Austral de Chile Animal Care Committee (No. 269/2016). Informed written owner consent was obtained for all cats included in the study.

### Study design

The sampling size was statistically estimated by assuming an expected prevalence of urinary shedding of pathogenic *Leptospira* of 4% with a 95% confidence interval (CI) of the estimation with a 5% precision, indicating that at least 236 cats had to be included. Between October 2016 and March 2017, cats of rural and urban origin from the Los Ríos and Los Lagos region in Southern Chile were included, trying to cover a geographic area in which a high antibody prevalence in cats had been reported previously [18].

### Cats

Included cats were patients of the small animal hospital of the Universidad Austral de Chile in Valdivia presented for health issues or for elective procedures (e.g. neutering, vaccination), client-owned cats participating in castration projects offered by the city of Valdivia or other communities, and cats presented by their owners specifically for study enrolement. Cats were offered one free vaccination and deworming as compensation for participating, if this was clinically justifiable. Cats were excluded from the study if they had received antimicrobial treatment within the last four weeks before sampling due to a decreased likelihood for detection of *Leptospira* spp. even if they were infected, or if they were living exclusively indoors or had only access to a balcony because of a low risk for infection.

### Sampling

Urine samples were collected via cystocentesis using a 21 G needle attached to 5 ml syringe. Cystocentesis was performed with palpation of the urinary bladder or ultrasound-guided. At the time of sampling, a physical examination was performed in each cat. A questionnaire with information about the cat and potential risk factors was answered by the owner. It included questions about signalment, living conditions, consumption of raw meat, drinking out of puddles, contact with rodents or eating rodents, and contact with other cats, dogs, or livestock and vaccination status.

### Detection of pathogenic *Leptospira* spp. by qPCR and culture

To increase the overall diagnostic sensitivity for the detection of pathogenic *Leptospira* spp. in urine, four diagnostic tools were used in each sample: *lipL32* quantitative qPCR, immunomagnetic-separation (IMS)-coupled with *lipL32* qPCR (IMS-qPCR), culture for *Leptospira* spp., and IMS-coupled with culture (Fig 1).

**Quantitative real-time *lipL32* polymerase chain reaction.** For each sample, at least 1 mL was centrifuged at 6,500 rpm for 15 minutes. The pellet was resuspended in 1 mL of 1X phosphate buffered saline (PBS) [137 mM NaCl, 2.7 mM KCl, 4.3 mM $Na_2HPO_4$, 1.4 mM $KH_2PO_4$ (pH 7)] and then transferred to a 1.5 mL microcentrifuge tube and re-centrifuged at 11,000 rpm for 5 minutes. Finally, the supernatant was discarded and the pellet was resuspended in 1mL of 1X PBS [26]. After this first cleaning and purification step, 100 μL were separated for a pretreatment step of immune-separation (see below) before proceeding with DNA extraction and direct qPCR. The rest of the urine was subjected to a DNA extraction-purification protocol using the High Pure PCR Template Preparation kit (Roche), following the manufacturer instructions.

The DNA templates obtained were analyzed in a qPCR system (Roche LightCycler 2.0), using a TaqMan probe and targeting the *lipL32* gene which is specific only for pathogenic *Leptospira* spp. [26]. The amplification mixture for each sample included 0.7 μM primers, 0.15 μM probe, 10 μL Master Mix TaqMan universal (Roche) and 5 μL DNA template, in a total volume of 20 μL. Samples were amplified with the following program: Initial denaturation at 95°C for 2 min, followed by 40 cycles of denaturation for 5 s at 95°C for 5 and annealing/elongation for 30 s at 58°C [26, 27]. The PCR system included a negative and positive control in order to survey the proficiency of the reaction as well as DNA extraction-negative and -positive controls. The positive control for both, DNA extraction and PCR protocols, corresponded to a pure culture of pathogenic *Leptospira* spp. in two dilutions with a known concentration of leptospires ($10^4$/mL and $10^2$/mL), where the two dilutions were used. Amplification efficiency (E) was calculated from the slope of the standard curve in each run using the following formula ($E = 10^{-1/slope}$).

**Immunomagnetic-separation (IMS)-coupled *lipL32* quantitative polymerase chain reaction.** The urine sample aliquot was pre-treated with an IMS method published previously [28], followed by a DNA extraction purification and molecular confirmation qPCR. The IMS-qPCR consisted of 4 steps (*in silico* peptide production; polyclonal antibody production; coating magnetic beads with polyclonal antibodies, and IMS–qPCR *Leptospira* spp. detection). The magnetic separation of pathogenic *Leptospira* spp. from urine samples and the subsequent washing steps were carried out using the automated BeadRetrieverTM System (Invitrogen Life Technologies, Grand Island, NY). Pathogenic *Leptospira* spp. were selectively concentrated from 0.5 mL processed samples obtained from the coated magnetic beads, as described above. The final product was suspended in 0.5 mL PBS for DNA extraction purification followed by qPCR for pathogenic *Leptospira* spp. as described above.

**Direct culture.** Two dilutions of urine (1:10 and 1:100) with 5 ml of liquid Ellinghausen-McCullough-Johnson-Harris (EMJH) medium supplemeted with 5-fluorouracil (0.2 mg/ml) were prepared immediately after urine sampling. Samples were cultured at 29°C for 13 weeks according to previous recommendations [29]. In this liquid medium, growth was defined as turbidity that becomes apparent at an estimated bacterial density of $1x10^8$ bacteria/mL. Each sample was observed for evidence of growth using dark-field microscopy for observation of viable, motile *Leptospira* spp. bacteria. DNA of positive isolations was used for molecular characterization using the above described extraction protocol.

**Immunomagnetic separation coupled to culture.** After immunomagnetic separation as described above, 100 μL of the final product were transferred into 5 ml EMJH medium and cultured at 29°C for 13 weeks. Each sample was observed for evidence of colony growth, as defined above. DNA of positive cultures was used for molecular characterization.

**Estimation of pathogenic *Leptospira* spp. shedding load.** Pathogenic *Leptospira* spp. cell numbers (genome equivalents) were estimated from either of the detection methods. Estimation was based on the concentration of *Leptospira* spp. DNA, obtained from positive samples,

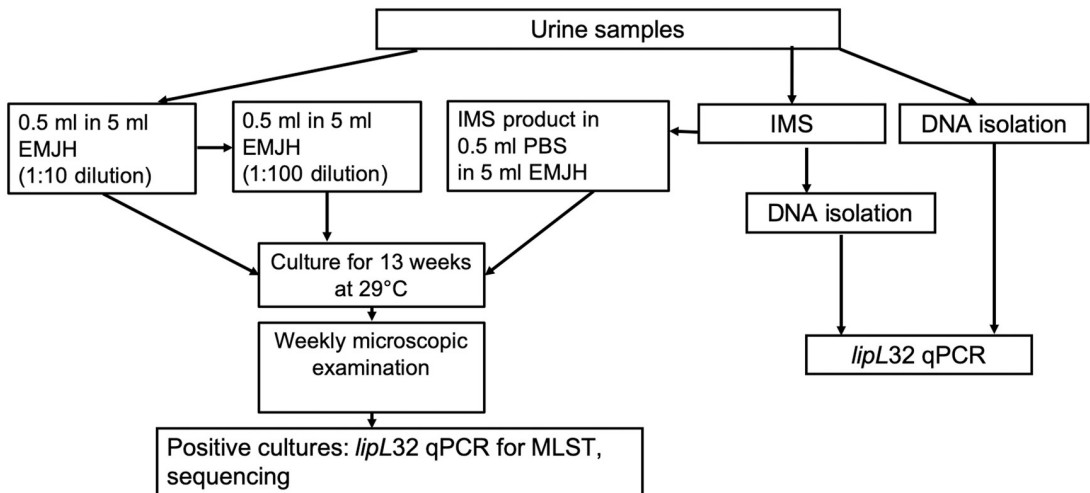

IMS = immunomagnetic separation, EMJH = Ellinghausen-McCullough-Johnson-Harris
medium, MLST = multilocus sequence typing, PBS = phosphate buffered saline

**Fig 1. Methods applied for identification of urinary shedding of pathogenic *Leptospira* spp.**

that was measured in a Nanoquant spectrophotometer (TECAN group, Männedorf, Schweiz) adjusted for a $10^6$ dilution. With the reference of the molecular weight of the genome of *Leptospira interrogans* serovar Hardjo-prajitno (GenBank accession number EU357983.1) a standard curve for estimation of pathogenic *Leptospira* spp. numbers in urine using a Roche 2.0 real-time PCR, according to the following equation was established [30, 28]:

$$\text{Genome equivalent} = \frac{\text{DNA concentration (ng/μl)} \times (6.022 \times 10^{23}\text{mol}^{-1})}{(4.659 \times 10^6 \text{ base pairs}) \times (1 \times 10^9 \text{ng/g} \times 660 \text{g/mol})}$$
$$\text{(accesion number EU357983.1)} \quad \text{(base mass)}$$

**Molecular characterization by multilocus sequence typing (MLST).** The "Multi Locus Sequence Typing" (MLST) method was used to molecularly characterize strains obtained from positive cultures. MLST was performed according to a protocol described by Thaipadungpanit et al. (2007) with the following housekeeping genes: *glmU*, *pntA*, *sucA*, *tpiA*, *pfkB*, *mreA*, and *caiB* [31]. Briefly, PCR amplification was carried out using *Leptospira* spp. DNA obtained from positive urine cultures, where the initial denaturation step was 94˚C for 5 min, followed by 35 cycles of 94˚C for 30 sec, annealing/elongation was at 50˚C for 60 sec for all genes, extension was at 72˚C for 50 sec, and then the final extension was at 72˚C for 7 min. The PCR products were purified using the Geneaid™ "Gene/DNA Genetic Extraction" kit, and sequencing was done in both directions with primers initially used for PCR amplification. Sequencing was performed using the sequencer kit "BigDye Terminator v. 3.1 cycle sequencing kit (ABI)" and the automated DNA sequencer "ABI Prism 3130xl Genetic". The sequences were analyzed using the Chrommas and Bioedits programs, and these were derived from the international database for free use (https://pubmlst.org/leptospira/) to obtain the allelic profile and to assign the sequence type (ST).

## Data analysis

The overall prevalence of urinary shedding (as detected by any of the 4 techniques) of pathogenic *Leptospira* spp. with their 95% confidence was calculated. Mann-Whitney test was used for comparison of *Leptospira* spp. loads between direct qPCR and IMS-qPCR. Statistical analysis to assess risk factors was performed by ordinary and mixed-logistic regression models using the location of the cat as a random slope effect to account for potential environmental heterogeneity.

The strategy for building the model consisted in obtaining unconditional models for initial screening of variables. Variables associated with the outcome (P<0.25) were eligible for inclusion in the conditional model that was built using a forward approach and a P value of 0.05 was used for interpreting the outcomes. Potential interactions were tested based on their biological significance. The best model was assessed by using the Bayesian information criterion (BIC) index as a measure of goodness-of-fit. Two analysis were performed: a) one for cats positive with any of the 4 techniques, combined as "*Leptospira*-shedding" cats, and b) one for "viable *Leptospira*-shedding" cats (culture- and IMS-coupled culture-positive cats). Analysis was done with lme4 package [32] of R (R Development Core Team (2008)) [33]. A power *post hoc* analysis was performed using WebPower package [34] for a evaluation of variables that were statistically not significant in the model.

## Results

### Shedding of pathogenic *Leptospira* spp.

In total, 231 cats from 11 locations at 7 municipal districts were included into the study (Fig 2). Overall, 30 cats were PCR-positive (Table 1), and 7 cats were culture-positive.

All 18 cats that were positive on qPCR were also positive on IMS-qPCR (Table 2). IMS-qPCR identified 12 additional cats as shedders of pathogenic *Leptospira* spp. Seven cats were positive in culture, 3 cats in direct culture and 4 other cats in IMS-coupled culture. None of them was positive in both culture methods. One of the cats that was positive in IMS-coupled culture was also positive in conventional qPCR and on IMS-qPCR. The other 6 culture-positive cats were PCR-negative. Individual data on signalment and health status are illustrated in Table 2. Data of geographic distribution of positive cats are combined in Table 3.

Multi locus sequence typing of isolates obtained of 7 positive cultures revealed that sequences of 6 cats were identical to available sequences of *L. interrogans* sequence type (ST) 17, and the sequence of one cat was identical to available sequences of *L. interrogans* ST 27 (Table 4).

### Risk factor analyses

For unconditional model (Table 5), health status (P<0.001), eating rodents (P = 0.08), reproductive status (P = 0.11), previous vaccination (P = 0.11), and age >1 year (P = 0.12) were variables primary associated with leptospiruria, and used for building a conditional model for cats that were PCR- and/or culture positive.

Conditional logistic regression after evaluating potential interactions and confounders is illustrated in Table 6 and included four risk factors. Two of them (being sick and being vaccinated) were significantly associated with shedding of pathogenic *Leptospira* spp. while the other two (age >1year and eating rodents) were not. The random intercept was statistically significant and accounted for 22.9% of the extra variability due to the place where the cat lived. The power estimation for this model was 90% which is considered high enough.

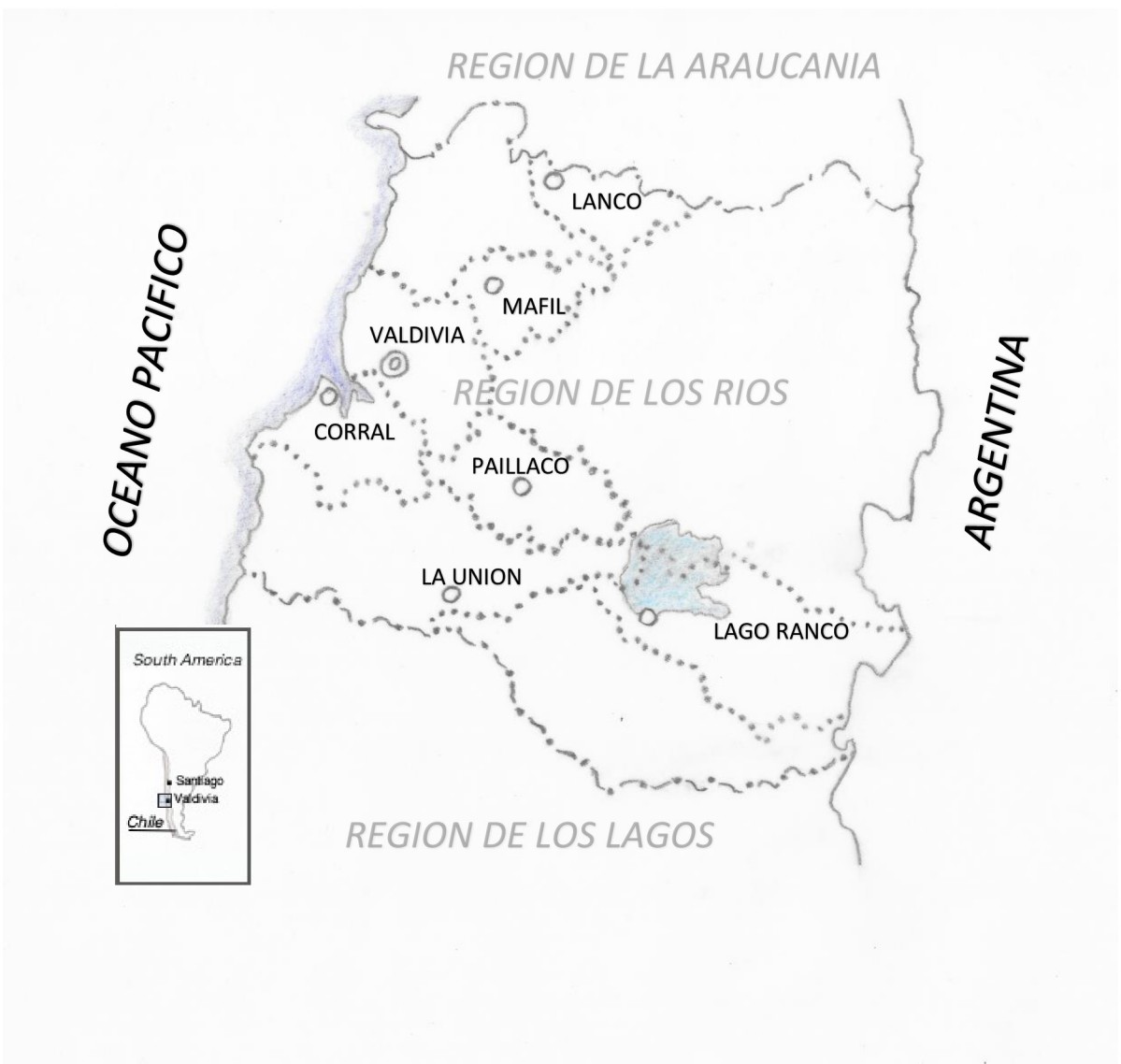

**Fig 2. Map of Latin America and of the region Los Rios in Southern Chile.** Sampled cats originated from the 7 delineated municipal districts.

**Table 1. Percentage and 95% Confidence Intervals (CI) of positive results from 231 urine samples using quantitative real-time PCR (qPCR), immunomagnetic-separation-coupled qPCR (IMS-qPCR), direct culture and IMS-coupled culture for identification of cats shedding pathogenic *Leptospira* spp. in urine.**

| Assay | Percentage (number) of positive samples | | Estimated bacterial load/mL | |
|---|---|---|---|---|
| | % (n) | 95% CI | Mean | 95% CI |
| qPCR | 7.8 (18) | 4.9–12.1 | 32.7 | 17–49 |
| IMS-qPCR | 13.0 (30) | 9.2–18.0 | 1,683,123 | 578,206–2,788,040 |
| Direct culture | 1.3 (3) | 0.3–3.9 | n. d. | n. d. |
| IMS-coupled culture | 1.7 (4) | 0.5–4.5 | n. d. | n. d. |
| Total | 15.6 (36) | 11.4–20.9 | n. d. | n. d. |

CI = confidence interval; n. d. = not determined

**Table 2. Signalment, origin and health status of cats with positive results on qPCR, immunomagnetic-separation (IMS)-coupled PCR (IMS-qPCR), direct culture and/or IMS-coupled culture and respective bacterial concentrations in number of genomic equivalents/mL (GE/ml).**

| Cat | Sex | Age (years) | Location | Status | Health status | qPCR | GE/ml estimated by qPCR | IMS-qPCR | GE/mL estimated by IMS-qPCR | Culture | IMS-coupled culture |
|---|---|---|---|---|---|---|---|---|---|---|---|
| 2 | fs | 1,5 | Valdivia | priv. | ear mites, chronically ill* | neg | 0 | neg | 0 | neg | **pos** |
| 8 | fs | 5 | Valdivia | priv. | healthy | neg | 0 | **pos** | 119 | neg | neg |
| 10 | f | 1,75 | Valdivia | priv. | healthy | **pos** | 35 | **pos** | 7,160,000 | neg | neg |
| 11 | f | 1 | Valdivia | priv. | healthy | **pos** | 26 | **pos** | 330,000 | neg | neg |
| 12 | f | 0,75 | Valdivia | priv. | healthy | **pos** | 96 | **pos** | 3,380,000 | neg | neg |
| 13 | f | 4 | Valdivia | priv. | healthy | **pos** | 42 | **pos** | 8,380,000 | neg | neg |
| 18 | f | 1 | Paillaco | priv. | chronically ill* | **pos** | 19 | **pos** | 124,800 | neg | neg |
| 22 | f | NA | Lago Ranco | stray cat | healthy | **pos** | 3 | **pos** | 2,680 | neg | neg |
| 25 | fs | 2 | Valdivia | priv. | unilateral nasal discharge | **pos** | 22 | **pos** | 9,440,000 | neg | neg |
| 26 | mc | 3 | Valdivia | priv. | hematuria | **pos** | 16 | **pos** | 632,000 | neg | **pos** |
| 29 | mc | 2 | Valdivia | priv. | acute renal failure | **pos** | 12 | **pos** | 6,840,000 | neg | neg |
| 30 | m | 2 | Valdivia | priv. | ear mites chronically ill* | neg | 0 | **pos** | 1,178 | neg | neg |
| 34 | f | 1 | Valdivia | priv. | healthy | **pos** | 124 | **pos** | 562,000 | neg | neg |
| 38 | f | 1 | Lanco | priv. | healthy | neg | 0 | **pos** | 630 | neg | neg |
| 40 | mc | 8 | Lanco | priv. | healthy | neg | 0 | **pos** | 3,020 | neg | neg |
| 43 | f | 6 | Lanco | priv. | healthy | neg | 0 | **pos** | 1,966 | neg | neg |
| 45 | f | 1 | Valdivia | priv. | healthy | **pos** | 62 | **pos** | 6,800,000 | neg | neg |
| 47 | f | 0,6 | Valdivia | priv. | stomatitis | neg | 0 | **pos** | 14,720 | neg | neg |
| 48 | m | 1 | Valdivia | priv. | stomatitis | neg | 0 | **pos** | 16,600 | neg | neg |
| 49 | f | 1,5 | Valdivia | priv. | healthy | neg | 0 | **pos** | 566 | neg | neg |
| 50 | mc | 4 | Valdivia | priv. | idiopathic cystitis, urethral stricture | neg | 0 | **pos** | 1,804 | neg | neg |
| 56 | m | 8 | Valdivia | priv. | healthy | **pos** | 14 | **pos** | 4,720,000 | neg | neg |
| 65 | fs | 4 | Valdivia | priv. | healthy | neg | 0 | neg | 0 | neg | **pos** |
| 66 | f | 1 | Valdivia | priv. | healthy | neg | 0 | neg | 0 | **pos** | neg |
| 67 | f | 3 | Valdivia | priv. | vomiting | neg | 0 | neg | 0 | **pos** | neg |
| 68 | mc | 10 | Valdivia | priv. | oral tumor | **pos** | 8 | **pos** | 352,000 | neg | neg |
| 69 | m | 6 | Valdivia | priv. | mammary tumors | **pos** | 16 | **pos** | 314,000 | neg | neg |
| 71 | m | NA | Mafil | priv. | healthy | **pos** | 14 | **pos** | 402,000 | neg | neg |
| 76 | m | 1 | Corral | priv. | healthy | **pos** | 45 | **pos** | 354,000 | neg | neg |
| 90 | mc | 5 | Valdivia | priv. | HCM | **pos** | 19 | **pos** | 394,000 | neg | neg |
| 91 | m | 4 | Valdivia | priv. | healthy | neg | 0 | neg | 0 | neg | **pos** |
| 121 | f | 0,4 | Valdivia | priv. | healthy | neg | 0 | **pos** | 5 | neg | neg |
| 130 | fs | 6 | Valdivia | priv. | healthy | **pos** | 14 | **pos** | 262,000 | neg | neg |
| 181 | f | 1,5 | Valdivia | priv. | healthy | neg | 0 | **pos** | 3,480 | neg | neg |
| 182 | f | 1,5 | Valdivia | priv. | healthy | neg | 0 | **pos** | 132 | neg | neg |
| 209 | m | 1,5 | Valdivia | priv. | healthy | neg | 0 | neg | 0 | **pos** | neg |

f = female, fs = female spayed, HCM = hypertrophic cardiomyopathy, m = male, mc = male castrated, NA = information not available, neg = negative, pos = positive,

priv. = privately owned; bold = positive in the respective test;

*Cats had the habitus of a chronically ill animal presenting with a reduced body condition score, dull fur and lethargy;

**Table 3. Description of sampling locations, number of cats sampled and number of cats positive in quantitative real-time PCR (qPCR) and/or immunomagnetic-separation (IMS)-coupled qPCR, as well as in direct culture and/or IMS-coupled culture.**

| Sampling location (Municipality) | All sampled cats n | PCR-positive cats n (%) | Culture-positive cats n (%) | All positive cats n (%) |
|---|---|---|---|---|
| Valdivia | 186 | 23 (12.4) | 7 (3.8) | 29 (15.6) |
| Corral | 12 | 1 (8.3) | 0 | 1 (8.3) |
| Paillaco | 10 | 1 (10.0) | 0 | 0 |
| Lanco | 7 | 3 (42.9) | 0 | 3 (42.9) |
| La Unión | 6 | 0 | 0 | 0 |
| Mafil | 5 | 1 (20.0) | 0 | 1 (20.0) |
| Lago Ranco | 5 | 1 (20.0) | 0 | 1 (20.0) |
| **Total** | **231** | **30 (13.0)** | **7 (3.0)** | **36** |

In addition, another model was used to evaluate potential risk factors associated with shedding of viable pathogenic *Leptospira* spp. (only cats positive on direct culture or IMS-coupled culture). For the unconditional models, only 3 variables were potential candidates: age >1 year, the reproductive status of the cat, and eating rodents were significantly associated with *Leptospira* spp.-positive urine cultures.

Although a mixed model was also assessed, comparison of the goodness-of-the-fit between both indicated that the mixed-model did not fit better. Therefore, ordinary logistic regression model results are shown. The conditional logistic regression model (Table 7) included 2 variables (age >1 year and eating rodents) that were significantly associated with shedding of viable pathogenic *Leptospira* spp.

## Discussion

Overall, 15.6% of cats included in the study were identified as shedders of pathogenic *Leptospira* spp.. This is considerably higher than the shedding rates from 0.8% to 4.9% in Thailand, Spain, Germany, Canada, and Malaysia [12, 14–16, 20]. One possible explanation might be the diagnostic techniques used in the present study as the application of an additional *Leptospira* spp. concentration technique (IMS-qPCR) increased the proportion of *Leptospira* spp.-shedding cats identified by PCR from 7.8% to 13.0% (12 additional cats). With direct culture and IMS-coupled culture another 6 shedding cats were identified. Other possible explanations could be related to the humid climatic conditions in the area where the study was performed with very high annual precipitations and the types of pathogenic *Leptospira* spp. circulating in the study area. In the present study, serovar identification was not performed, but the culture-grown leptospires could be characterized as *L. interrogans* ST 17 and ST 27 (pathogenic Subgroup 1) based on MLST. Other studies using whole-genome sequencing provided evidence that strains of pathogenic leptospires differ in genetic features and virulence factors for

**Table 4. Multi locus sequence typing of PCR products, allelic profile, and sequence types (ST) from 7 *Leptospira* spp.-positive urine cultures.**

| Cat # | *caiB* | *glmU* | *mreA* | *pfkB* | *pntA* | *sucA* | *tpiA* | ST | Product |
|---|---|---|---|---|---|---|---|---|---|
| 2 | 8 | 11 | 4 | 10 | 1 | 21 | 2 | 17 | *L. interrogans* Icterohaemorrhagiae Copenhageni |
| 26 | 8 | 11 | 4 | 10 | 2 | 21 | 65 | 17 | *L. interrogans* Icterohaemorrhagiae Copenhageni |
| 65 | 8 | 4 | 4 | 10 | 2 | 21 | 2 | 17 | *L. interrogans* Icterohaemorrhagiae Copenhageni |
| 66 | 19 | 11 | 4 | 10 | 1 | 21 | 65 | 17 | *L. interrogans* Icterohaemorrhagiae Copenhageni |
| 67 | 8 | 4 | 4 | 10 | 2 | 21 | 2 | 17 | *L. interrogans* Icterohaemorrhagiae Copenhageni |
| 91 | 8 | 4 | 4 | 10 | 1 | 21 | 2 | 17 | *L. interrogans* Icterohaemorrhagiae Copenhageni |
| 209 | 19 | 11 | 13 | 10 | 12 | 3 | 3 | 27 | *L. interrogans* Autumnalis Autumnalis |

**Table 5. Unconditional logistic regression model results for pathogenic *Leptospira* spp. shedding cats (PCR- and/or culture-positive cats).**

| Variable | Category | n | *Leptospira* spp. shedding | | Odds ratio | 95% CI |
|---|---|---|---|---|---|---|
| | | | **Positive** | **Negative** | | |
| | | | **n (%)** | **n (%)** | | |
| **Age** | ≤1 year | 98 | 11 (11.22) | 87 (88.78) | Ref. | |
| | >1 year | 112 | 23 (20.54) | 89 (79.46) | 1.8* | 0.86–3.90 |
| **Gender** | Female | 138 | 22 (15.94) | 116 (84.06) | Ref. | |
| | Male | 89 | 14 (15.73) | 75 (84.27) | 0.98 | 0.46–2.02 |
| **Reproductive status** | Neutered | 52 | 12 (23.07) | 40 (76.92) | Ref. | |
| | Intact | 175 | 24 (13.71) | 151(86.29) | 0.53* | 0.25–1.15 |
| **Origin** | Rural | 81 | 15 (18.51) | 66 (81.48) | Ref. | |
| | Urban | 144 | 20 (13.89) | 124 (86.11) | 0.70 | 0.34–1.47 |
| **Ownership** | Feral | 11 | 2 (18.18) | 9 (81.82) | Ref. | |
| | Privately owned | 213 | 33 (15.49) | 180 (84.51) | 0.83 | 0.20–5.63 |
| **Weight** | >3kg | 99 | 14 (14.14) | 85 (85.86) | Ref. | |
| | ≤3 kg | 96 | 11 (11.46) | 85 (88.54) | 1.22 | 0.79–1.86 |
| **Health status** | Healthy | 208 | 23(11.06) | 185 (88.94) | Ref. | |
| | Sick | 23 | 13 (56.52) | 10 (43.48) | 4.12* | 1.81–9.23 |
| **Drinking out of puddles** | No | 159 | 25 (15.72) | 134 (84.28) | Ref. | |
| | Yes | 66 | 10 (15.15) | 56 (84.85) | 0.97 | 0.42–2.10 |
| **Contact with rodents** | No | 103 | 14 (13.59) | 89 (84.41) | Ref. | |
| | Yes | 122 | 21 (17.21) | 101 (82.79) | 1.35 | 0.65–2.86 |
| **Eating rodents** | No | 170 | 22 (12.94) | 148 (87.06) | Ref. | |
| | Yes | 43 | 10 (23.26) | 33 (76.74) | 2.08* | 0.87–4.75 |
| **Consumption of raw meat** | No | 163 | 28 (17.18) | 135 (82.82) | Ref. | |
| | Yes | 53 | 6 (11.32) | 47 (88.68) | 0.64 | 0.24–1.58 |
| **Contact with other cats** | No | 12 | 2 (16.67) | 10 (83.33) | Ref. | |
| | Yes | 211 | 33 (15.64) | 178 (84.36) | 1.07 | 0.38–2.67 |
| **Contact with dogs** | No | 49 | 8 (16.33) | 41 (83.67) | Ref. | |
| | Yes | 173 | 27 (15.61) | 146 (84.39) | 0.96 | 0.14–3.78 |
| **Contact with livestock** | No | 172 | 29 (16.86) | 143 (83.12) | Ref. | |
| | Yes | 52 | 6 (11.54) | 46 (88.46) | 0.643 | 0.25–1.65 |
| **Previous vaccinations#** | No | 121 | 13 (10.7) | 108 (89.3) | Ref. | |
| | Yes | 44 | 9 (20.4) | 35 (79.6) | 2.13* | 0.82–5.39 |

*P<0.25; CI = Confidence interval, Ref. = Reference category

# vaccinations against panleukopenia, feline herpesvirus and calicivirus and rabies in variable combinations and intervals

persistence outside the host, during entry into the host, as well as for colonization and persistence within the host organism [35, 36], suggesting that not all pathogenic *Leptospira* spp. are equally virulent [37]. Therefore, it cannot be excluded that the identified strains in the present study might have a higher virulence or be better adapted to invade the species cat than strains in other geographic regions.

There is only one study with a much higher proportion (68%) of shedding cats in Taiwan [11]. However, this study was performed after a typhoon with an associated outbreak of leptospirosis in humans [38] and included a high proportion of stray cats (68%), whereas in the present study most of the cats were privately owned well-fed cats, and only 31% of them had been observed by their owners eating rodents. In addition, in the study from Taiwan, a

**Table 6. Conditional logistic regression model results for pathogenic *Leptospira* spp. shedding cats (PCR- and/or culture-positive cats).**

| Variable | Category | Odds ratio | 90% CI |
|---|---|---|---|
| Age | >1 year | 1.36 | 0.54–3.42 |
| | ≤1 year | Ref. | |
| Health status | Healthy | Ref. | |
| | Sick | 3.04* | 1.10–8.39 |
| Eating rodents | No | Ref. | |
| | Yes | 1.20 | 0.40–3.55 |
| Previous vaccinations# | No | Ref. | |
| | Yes | 2.93* | 1.18–7.24 |

*P<0.05; Bayesian information criterion index = 144.9

CI = confidence interval, Ref. = Reference category

# vaccinations against panleukopenia, feline herpesvirus and calicivirus and rabies in variable combinations and intervals

different PCR technique (not a quantitative PCR) was applied and not all primers used in that study were specific for pathogenic *Leptospira* spp.

While PCR can detect DNA of viable and killed bacteria, culture is only positive when a sufficient number of viable *Leptospira* spp. is present. Recently, a study from Malaysia demonstrated cultivation of pathogenic *Leptospira* spp. from renal tissue in 3/82 clinically healthy shelter cats and from both, renal tissue and urine, in 1/82 of these cats [20]. In addition to that study, there is one case report of a cat with a positive *Leptospira* spp. culture originating from the same geographic area as the present study [10]. The present study also yielded positive *Leptospira* spp. culture results from feline urine. Feline urine with its higher osmolality compared to dogs and humans [39] is a hostile environment for bacterial growth and therefore could prohibit the shedding of viable *Leptospira* spp. in many cats. This might explain the low proportion of culture-positive cats in the present study. Low pH is another physicochemical property of the urine contributing to the defense system against bacteria [40] and *Leptospira* spp. do not survive well in acidic urine but remain viable in alkaline urine [41]. However, in the present study, the mean urine pH was identical in culture-positive and in culture-negative cats.

Usage of magnetic separation enhances the analytical specificity and sensitivity of the subsequent detection method, given that it helps to remove the presence of many inhibitory substances or other organisms [42]. Still, only 7 cats (3.0%) were culture-positive (3 direct culture, 4 IMS-coupled culture) compared to 30 (13.0%) PCR-positive cats. However, detection of positive cultures nevertheless demonstrates that feline urine can be a source of possible

**Table 7. Conditional logistic regression model results for cats with positive urine cultures (direct culture and/or IMS-coupled culture) for *Leptospira* spp.**

| Variable | Category | Odds ratio | 90% CI |
|---|---|---|---|
| Age | >1 year | 6.61 | 1.08–40.40 |
| | ≤1 year | Ref. | |
| Eating rodents | No | Ref. | |
| | Yes | 3.87 | 1.03–14.49 |

CI = confidence interval, Ref. = Reference category

contamination of the environment with viable *Leptospira* spp. One possible explanation for the discrepancy in the results of direct and IMS-coupled cultures is the hypothesis that antibodies used in IMS could have a neutralizing effect for the growth of *Leptospira* spp. in culture.

In the present study, only 1 of the 7 culture-positive cats was also PCR-positive (qPCR and IMS-qPCR). One possible reason for the negative PCR in the culture-positive cats could be the presence of PCR inhibitors. In biologic fluids, such as urine, presence of PCR inhibitors has been reported to cause false negative PCR results [43]. However, positive controls in the present study showed hardly or no inhibition. Interestingly, a similar discrepancy was seen in the only other study in cats reporting the isolation of *Leptospira* spp. from feline kidneys and urine [20]. In that study, all 4 culture-positive cats had negative urine PCR results. This indicates that there might be substances/chemicals present in the PCR processing but not in the samples that are cultured that could be inhibitory for certain steps in the PCR. Other reasons might also be the level of detection on qPCR and possibly the selection of primers that do not bind to all DNA targets of *Leptospira* spp. However, molecular characterization of isolates later on revealed sequence types that should have been recognized by PCR.

In the present study, serovar identification was not performed. Nevertheless, it was possible to molecularly characterize isolates via MLST, and identified sequences were assigned to ST 17 and ST 27, which corresponds to *L. interrogans (Pathogenic Leptospira Subgroup 1)* [44]. Although the Taxonomic Subcommittee does not yet accept MLST for serovar identification, the information reported from the MLST database used in the analysis of the isolated strain in this study, reports the ST 17 as *Leptospira interrogans* serogroup Icterohaemorrhagiae serovar Copenhageni, and the ST 27 as *L. interrogans* serogroup Autumnalis serovar Autumnalis. Previously characterized *Leptospira* spp. isolates from cats´ kidney or urine had high similarity to *Leptospira interrogans* serovar Pomona (1 cat from this area of Chile) [10] or to *Leptospira interrogans* serovar Bataviae (4 cats from Malaysia) [20]. Whereas, *Leptospira interrogans* Bataviae is reported to be the most common isolated serovar in humans, dogs and rats in Malaysia [45–47], *L. interrogans* serovar Autumnalis is among the most commonly reported serovars in dogs and cats in Chile [18, 48, 49], and antibodies against *L. interrogans* Icterohaemorrhagiae have previously been reported in rodents in this area [50].

Four factors (age (≤1 year/>1year), health status, eating rodents, and vaccination status) were present in the final conditional model for all *Leptospira* spp. shedding cats (cats positive in any of the 4 techniques), and 2 factors (age (≤1 year/>1year), eating rodents) for the "viable *Leptospira* spp. shedding" cats (culture- or IMS-coupled culture-positive cats).

Adult stage (>1 year) was significantly associated with leptospiruria (odds ratio 6.8, CI 90% 1.08–40,4). Previous studies already reported older age as a risk factor for *Leptospira* spp. infection [15, 51]. A study in Thailand identified age >4 years as the only risk factor for infection (urine qPCR-positive status and/or serum antibodies in MAT) in a multivariate analysis [15]. A possible explanation for this is longer exposure to infectious environmental sources and therefore a higher risk of being infected. It is not known how long cats remain carriers after infection or if and under which circumstances they can eliminate the infection. Risk factors for *Leptospira* spp. infection have been investigated in several studies. Contact of cats with livestock animals was significantly associated with presence of *Leptospira* spp. antibodies in a study performed in Chile [18] but such an association between shedding and contact with livestock animals was not identified in the present study.

The proportion of sick cats was significantly higher in *Leptospira* spp. shedding cats (36,1%) than in non-shedding cats (5,1%). Clinical cases of leptospirosis in cats are rarely described. In one case series with 3 cats and 2 further reports of single cases, cats presented with acute onset of polydipsia and polyuria, lethargy, anorexia, and hematuria, and had various degrees of kidney disease or increased liver enzymes [8–10]. In a more recent case report, a cat

presented with a 2-months history of hematuria, had no abnormalities in serum chemistry but a marked leukocytosis [10]. One leptospiruric cat (positive in qPCR and IMS-qPCR) of the present study also suffered from acute renal failure (history of acute lethargy and anorexia, severe renal azotemia), and 2 presented with hematuria (1 cat positive in qPCR, IMS-qPCR and IMS-coupled culture, 1 cat positive in IMS-qPCR). These clinical signs could indeed be associated with leptospirosis and represent acute infection. In the other 10 sick cats with different organs affected, a causal relationship of infection with *Leptospira* spp. is not likely. However, the higher proportion of sick cats shedding pathogenic *Leptospira* spp. in the present study suggests that *Leptospira* spp. could be one of the etiological factors in the development of these clinical problems, and that diagnostic testing *Leptospira* spp. infection should also be considered in cats with relevant clinical signs. Alternatively, it could be discussed that sick cats might be immunosuppressed and that an impaired immune status increases the risk of leptospiral infection and subsequent shedding. This, however, has not been reported so far. On the other hand, sick cats shedding pathogenic *Leptospira* spp. could represent chronically infected individuals. It would have been interesting to further monitor these cats to see if they remained chronic shedders.

Eating rodents was found to be associated with an increased risk of shedding viable pathogenic leptospires. In general, rodents are a natural reservoir pathogenic *Leptospira spp.*, and rats are reservoir hosts of the Icterohaemorrhagiae and Copenhageni serovars [52]. Therefore, prey–predator transmission between cats and rodents is possible by hunting and is believed to be the main source of infection in cats [12, 53]. In the contrary, a negative association between cat ownership and the presence of *Leptospira* antibodies in people in an urban population in North America was demonstrated, and cats appear to have a protective role by decreasing contact of people with rodents that act as reservoir hosts [54].

An unforeseen result was the statistically significant association of an increased risk of shedding pathogenic *Leptospira* spp. and previous vaccination against feline panleukopenia, feline herpes- and calicivirus infection and/or rabies. The reason for this association is unknown. Immunological host-pathogen processes that could affect shedding of leptospires by cats in any direction are conceivable but specific information is missing. Weather there is an interference in susceptibility of vaccinated cats to act as shedders of *Leptospira* spp. is an interesting aspect that should be further investigated.

Results of the present study show that cats can shed viable pathogenic *Leptospira* spp. and, therefore, could act as transmitters to humans and livestock. Results of PCR and culture with and without separation and concentration, taken together, showed that: i) the bacterial load in feline urine is relatively low; ii) in 90% of feline shedders it was not possible to culture the organism suggesting that viability of the bacteria in feline urine is negatively affected by the hostile environment, e.g. the relatively high osmolality of feline urine; and iii) separation and concentration followed by qPCR significantly increases the sensitivity of detection of leptospiral DNA in feline urine.

Apart from the possible negative effect on bacterial viability of the hostile environment that is provided by feline urine and the stage of disease, there is another potential cause of negative culture result. There is a known difficulty to isolate *Leptospira* spp. from infected hosts and get them growing *in vitro*. Despite the presence of viable leptospires in clinical samples of an infected host, it is in many cases very difficult to isolate the bacteria in media due to the *in vivo* phenotype of the host-derived bacteria and their lack of capacity to adapt to *in vitro* conditions. Based on this general problem with cultural isolation of *Leptospira* spp. from infected humans and animals, it can be concluded that, despite this difficulty, viable *Leptospira* spp. were detected in the urine of 7 cats, which demonstrates the zoonotic potential of this infection in cats.

## Conclusion

In this region of Southern Chile, 15.6% of cats were leptospiruric and 3.0% of cats were *Leptospira* spp. culture-positive. The present study is one of the first reporting growth of *Leptospira* spp. from feline urine in culture and describing genetic characteristics of culture-derived isolates. Even though feline urine appears to be a hostile environment for leptospires, shedding cats present a potential risk for zoonotic transmission.

## Supporting information

**S1 Questionnaire.**
(DOCX)

**S2 Questionnaire.**
(DOCX)

## Acknowledgments

We thank all cat owners and cats for participation in the study.

## Author Contributions

**Conceptualization:** Roswitha Dorsch, Miguel Salgado, Ananda Müller, Katrin Hartmann.

**Data curation:** Roswitha Dorsch, Javier Ojeda, Theo Eberhard.

**Formal analysis:** Roswitha Dorsch, Javier Ojeda, Gustavo Monti, Theo Eberhard.

**Funding acquisition:** Roswitha Dorsch, Gustavo Monti, Ananda Müller, Katrin Hartmann.

**Investigation:** Roswitha Dorsch, Javier Ojeda, Miguel Salgado, Bernadita Collado, Camillo Tomckowiack, Carlos Tejeda, Theo Eberhard.

**Methodology:** Miguel Salgado, Bernadita Collado, Camillo Tomckowiack, Carlos Tejeda, Ananda Müller.

**Project administration:** Roswitha Dorsch, Gustavo Monti.

**Resources:** Roswitha Dorsch, Javier Ojeda, Miguel Salgado.

**Software:** Roswitha Dorsch, Gustavo Monti.

**Supervision:** Roswitha Dorsch, Miguel Salgado.

**Validation:** Miguel Salgado, Gustavo Monti, Henricus L. B. M. Klaasen.

**Visualization:** Roswitha Dorsch, Theo Eberhard.

**Writing – original draft:** Roswitha Dorsch, Miguel Salgado, Gustavo Monti, Katrin Hartmann.

**Writing – review & editing:** Roswitha Dorsch, Javier Ojeda, Miguel Salgado, Gustavo Monti, Ananda Müller, Theo Eberhard, Henricus L. B. M. Klaasen, Katrin Hartmann.

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
