## [Decision Letter · Decision Letter 0]

23 Jul 2020

PONE-D-20-16002

Cats shedding pathogenic Leptospira spp. – an underestimated zoonotic risk?

PLOS ONE

Dear Dr. Dorsch,

Thank you for submitting your manuscript to PLOS ONE. After careful consideration, we feel that it has merit but does not fully meet PLOS ONE’s publication criteria as it currently stands. Therefore, we invite you to submit a revised version of the manuscript that addresses the points raised during the review process.

ACADEMIC EDITOR: Estimation of shedding load formula need appropriate reference support. Information about the pH of the urine sample give some clue to improvise the isolation methods.

We look forward to receiving your revised manuscript.

Kind regards,

Kalimuthusamy Natarajaseenivasan

Academic Editor

PLOS ONE

Journal Requirements:

2. In your Methods section, please provide additional details regarding participant consent from the owners of the animals. In the ethics statement in the Methods and online submission information, please ensure that you have specified (1) whether consent was informed and (2) what type you obtained (for instance, written or verbal). If the need for consent was waived by the ethics committee, please include this information.

4. In your Methods section, please state where the participants were recruited for your study.

5.We note that [Figure(s) 2] in your submission contain [map/satellite] images which may be copyrighted. All PLOS content is published under the Creative Commons Attribution License (CC BY 4.0), which means that the manuscript, images, and Supporting Information files will be freely available online, and any third party is permitted to access, download, copy, distribute, and use these materials in any way, even commercially, with proper attribution. For these reasons, we cannot publish previously copyrighted maps or satellite images created using proprietary data, such as Google software (Google Maps, Street View, and Earth). For more information, see our copyright guidelines: http://journals.plos.org/plosone/s/licenses-and-copyright.

1.    You may seek permission from the original copyright holder of Figure(s) [2] to publish the content specifically under the CC BY 4.0 license. 

6.Thank you for stating the following financial disclosure:

 [Bavarian Academic Center for Latin America supported the study by paying for travel expenses (between Valdivia and Munich on two occasions) for planing the study. The Dirección de Investigación y Desarollo of the Universidad Austral de Chile supported the study by covering part of the costs for the material necessary for sampling and part of additional costs, e.g. transport and compensation of owners for study participation. For compensation, all included cats were vaccinated and dewormed. MSD Animal health provided the resources for sampling, clinical investigation of cats and all laboratory investigations. H.L.B.M. Klaasen is an employee of a company (MSD Animal Health) developing and marketing animal vaccines and pharmaceuticals. He contributed in preparation of the the mansucript and had valuable input in interpretation of the data. ].               

7.Thank you for stating the following in the Competing Interests section:

[H.L.B.M. Klaasen is an employee of a company (MSD Animal Health) developing and marketing animal vaccines and pharmaceuticals. Otherwise the authors declare that they have no conflict of interest with respect to the research, authorship and/or publication of this article.].

Additional Editor Comments (if provided):

The qPCR technique using LipL32 need the reference support to know the probe details in the method section. Really wonder to notice that out of the 7 culture/IM culture positive cases just one turned to be positive for PCR and the justification of presence of PCR inhibitors in the urine need more authentication. If it is the case, then it reflects on the negative samples as false negatives due to the PCR inhibitors. Since they have used the high pure PCR template preparation kit and practically, they give high pure genomic DNA without any inhibitors. Authors need to explain it well in the discussion.

Estimation of shedding load formula need appropriate reference support.

Is there any information about the pH of the collected urine sample. That would give some idea to improvise the isolation frequency.

Reviewers' comments:

Reviewer's Responses to Questions

**Comments to the Author**

1. Is the manuscript technically sound, and do the data support the conclusions?

Reviewer #1: Yes

Reviewer #2: Partly

2. Has the statistical analysis been performed appropriately and rigorously? 

Reviewer #1: Yes

Reviewer #2: I Don't Know

3. Have the authors made all data underlying the findings in their manuscript fully available?

Reviewer #1: Yes

Reviewer #2: Yes

4. Is the manuscript presented in an intelligible fashion and written in standard English?

Reviewer #1: Yes

Reviewer #2: Yes

5. Review Comments to the Author

Reviewer #1: It is a well-structured document with scientific rigour. The description of the methodology should be improved, for another researcher to reproduce the experiments described.

For details, please see the attached document.

Reviewer #2: The purposes of this study were to determine the proportion of cats shedding pathogenic Leptospira in Southern Chile, to determine the molecular profile of the cultured isolates, and to identify possible risk factors regarding Leptospira spp. shedding.

I think the results of this paper are definitely very interesting and highlight the importance of further investigating leptospirosis in cats. However, there are still some issues need to be addressed before further consideration.

Abstract:

Line 25: Change the Leptospira to Leptospira spp.

Line 26: This sentence need to be rephrased as in the reference 19, it has been reported from urine and kidney.

Line 33: Immunomagnetic

Introduction:

Line 59: should be under investigated rather than rarely suffer.

Line 63: The proportions were also very much depending on samples size and laboratory techniques used. Prevalence is more commonly used. Suggest changing the proportions to prevalence.

Line 69-76: Suggest shortening the paragraph.

Line 84-86: This was the result obtained from previous publication, it will be more informative if authors can share about the importance of that finding.

Line 87: How big is the human population? Since this paper concerns about the role of cats in leptospirosis. Strongly suggest the authors to add in human leptospirosis in Chile in the introduction in order to highlight how severe is the problem in human and lead to the investigating in companion animal.

Material and methods:

Line 104: Does the locations and time of sampling affected the result?

Line 111: would suggest providing a justification for the inclusion for outdoor cat only in discussion part. The definition of outdoor need to be specified. Roam freely? Community cat?

Line 117: The physical examination was performed by who? Was there any consent from owner?

Line 124: MAT is always the gold standard in investigating leptospirosis. Is there any reason why it was not performed?

Line 123: There is lack of justification the usage of qPCR, IMS-qPCR, culture and IMS-coupled culture concurrently in this study. From the line 336 to 346, IMS-qPCR was reported to be more sensitive than qPCR from the previous study. Hence, is the comparison between the different techniques also one of the objectives of the study?

Table 2

Please be more specific about the chronically ill. Would like to know what type of illness.

Discussion:

Line 332: The recruited cats were not indoor, hence, how accurate the owner can observe about the rat-hunting and eating behaviour?

Line 349-351: Since it has been reported before, I will recommend removing “first one”.

Line 355-356: Need to discuss a little bit about the findings obtained from direct culture and IMS-coupled culture. Why there were certain samples positive with direct culture, some with IMS-coupled culture?

Line 424-425: it also affected by the stage pf the leptospirosis

Line 427: that should be one of the aims of the study

6. PLOS authors have the option to publish the peer review history of their article (what does this mean?). If published, this will include your full peer review and any attached files.

Reviewer #1: No

Reviewer #2: No

---

## [Author Response · Author response to Decision Letter 0]

21 Aug 2020

Please see Response to Reviewers letter at the end of this PDF

---

## [Decision Letter · Decision Letter 1]

17 Sep 2020

Cats shedding pathogenic Leptospira spp. – an underestimated zoonotic risk?

PONE-D-20-16002R1

Dear Dr. Dorsch,

We’re pleased to inform you that your manuscript has been judged scientifically suitable for publication and will be formally accepted for publication once it meets all outstanding technical requirements.

Kind regards,

Kalimuthusamy Natarajaseenivasan

Academic Editor

PLOS ONE

Additional Editor Comments (optional):

Reviewers' comments:

Reviewer's Responses to Questions

**Comments to the Author**

1. If the authors have adequately addressed your comments raised in a previous round of review and you feel that this manuscript is now acceptable for publication, you may indicate that here to bypass the “Comments to the Author” section, enter your conflict of interest statement in the “Confidential to Editor” section, and submit your "Accept" recommendation.

Reviewer #1: All comments have been addressed

Reviewer #2: All comments have been addressed

2. Is the manuscript technically sound, and do the data support the conclusions?

Reviewer #1: Yes

Reviewer #2: Yes

3. Has the statistical analysis been performed appropriately and rigorously? 

Reviewer #1: Yes

Reviewer #2: Yes

4. Have the authors made all data underlying the findings in their manuscript fully available?

Reviewer #1: Yes

Reviewer #2: Yes

5. Is the manuscript presented in an intelligible fashion and written in standard English?

Reviewer #1: Yes

Reviewer #2: Yes

6. Review Comments to the Author

Reviewer #1: you have done a great job, I am sure the results of your study help to improve the knowledge of leptospira infection in cats.

Just a small mistake on line 122 the word AND is repeated.

Reviewer #2: All the comments have been addressed and I truly enjoy reading this manuscript. This manuscript is important to bridge the current lacking information in the role of companion animals in leptospirosis.

7. PLOS authors have the option to publish the peer review history of their article (what does this mean?). If published, this will include your full peer review and any attached files.

Reviewer #1: No

Reviewer #2: No

---

## [Editor Report · Acceptance letter]

14 Oct 2020

PONE-D-20-16002R1 

Cats shedding pathogenic Leptospira spp. – an underestimated zoonotic risk? 

Dear Dr. Dorsch:

I'm pleased to inform you that your manuscript has been deemed suitable for publication in PLOS ONE. Congratulations! Your manuscript is now with our production department. 

Kind regards, 

on behalf of

Dr. Kalimuthusamy Natarajaseenivasan 

Academic Editor

PLOS ONE